# Revisiting OOD Generalization in Programmatic RL

**Amirhossein Rajabpour** [1 2]   **Kiarash Aghakasiri** [1 2]   **Sandra Zilles** [3 2]   **Levi H. S. Lelis** [1 2]

## Abstract

Programmatic policies are often reported to generalize better than neural policies in reinforcement learning (RL) benchmarks. We revisit some of these claims and show that much of the observed gap arises from uncontrolled experimental factors rather than intrinsic representational reasons. Re-evaluating three core benchmarks used in influential papers—TORCS, Karel, and Parking—we find that neural policies, when trained with a few modifications, such as sparse observations and cautious intrinsic reward functions, can match or exceed the out-of-distribution (OOD) generalization of programmatic policies. We argue that a representation enables OOD generalization if (i) the policy space it induces includes a generalizing policy and (ii) the search algorithm can find it. The neural and programmatic policies in prior work are comparable in OOD generalization because the domain-specific languages used induce policy spaces similar to those of neural networks, and our modifications help the gradient search find generalizing solutions. By disentangling representational factors from experimental confounds, we advance our understanding of what makes a representation succeed or fail at OOD generalization.

## 1. Introduction

Reinforcement learning (RL) has led to remarkable successes in domains ranging from games to robotics, largely by representing policies as highly parametrized neural networks (Mnih et al., 2015; Schulman et al., 2017; Lillicrap et al., 2019). However, neural policies often struggle to generalize outside the distribution of their training environ-

ments, exhibiting brittle behavior when confronted with out-of-distribution (OOD) scenarios. In contrast, a growing literature on policies expressed as computer programs presents empirical evidence of superior OOD generalization compared to neural representations (Verma et al., 2018; 2019; Trivedi et al., 2021; Inala et al., 2020). Despite this empirical evidence, the literature does not explain why programmatic representations generalize better than neural ones. Motivated to find an explanation, we re-evaluated the experiments of three influential works. Our re-evaluation focuses on OOD generalization, although programmatic representations can have important advantages over neural representations, such as interpretability (Kohler et al., 2024).

Our re-evaluation shows that the reported gap between neural and programmatic representations in generalization stems from uncontrolled experimental factors rather than representational differences. With a few adjustments to the training pipeline, neural policies generalized as well as programmatic ones. For example, in the TORCS experiments of (Verma et al., 2018), the neural models failed to generalize to unseen race tracks because they excelled in optimizing for the car's speed on the training track. Programmatic policies are less effective at optimizing speed and thus generalize better to tracks with sharper turns. Once we used a safer intrinsic reward function that de-emphasized speed, neural policies matched programmatic ones in generalization. Our re-evaluation still leaves open the question of whether programmatic representations can generalize better than the neural models used in previous work, and, if so, what factors would explain such an advantage.

We posit that a type of representation is successful in OOD generalization if the following two conditions are satisfied: (i) it encodes in its space a solution that generalizes (expressivity), and (ii) the search process used can find and return a solution that generalizes (discoverability). The languages used in the studies we re-evaluate define a space of solutions that is similar, if not identical, to the space of solutions a neural network induces. This means that, if the programmatic space had solutions that generalized, then the neural space also encoded such solutions. The changes that we made to the training pipeline in our re-evaluation allowed the gradient search to find the solutions that generalize.

The comparison of different representations for solving RL

[1]Department of Computing Science, University of Alberta, Edmonton, Canada. [2]Alberta Machine Intelligence Institute (Amii), Edmonton, Canada. [3]Department of Computer Science, University of Regina, Regina, Canada.. Correspondence to: Amirhossein Rajabpour <arajabpo@ualberta.ca>.

*Proceedings of the 43rd International Conference on Machine Learning*, Seoul, South Korea. PMLR 306, 2026. Copyright 2026 by the author(s).

problems can be examined considering expressivity and discoverability, either in conjunction or separately. We conjecture that previous work inadvertently evaluated programmatic and neural representations that satisfied expressivity, and discoverability was controlled for the search in the programmatic space, but not in the neural space. As demonstrated by our experiments, controlling for discoverability when both representations evaluated satisfy expressivity can be difficult. Given that expressivity is satisfied, the ability to retrieve solutions that generalize depends on search heuristics that vary with the domain and representation used.

Our results suggest that empirical evidence from previous work does not clearly separate differences in expressivity from differences in discoverability. Clarifying this distinction in future experiments can help in understanding OOD generalization results. Given our results, we argue that the discussion of representation learning in the context of OOD generalization should shift from the choice between programmatic and neural approaches to selecting representations that meet the computational requirements of the underlying problem. For example, if the underlying problem requires memory that scales with the problem's input size, then the representation should encode solutions whose memory usage scales with the input size, regardless of whether the representation is programmatic or neural. In our terminology, satisfying these computational requirements concerns expressivity: the representation must contain solutions that can generalize OOD. The remaining question is discoverability: whether the learning algorithm can find such solutions. We believe this shift in focus can help design novel representations that enable OOD generalization. We make our code and experimental setup publicly available at code.

## 2. Problem Definition

We consider sequential decision-making problems modeled as partially observable Markov decision processes (POMDPs) $\mathcal{M} = (S, A, O, p, \Omega, r, \mu, \gamma)$. Here, $S$ is a set of states, $A$ a set of actions, and $O$ a set of observations. The transition function $p : S \times A \to \Delta(S)$ and observation function $\Omega : S \times A \to \Delta(O)$ specify the environment dynamics and the observation process. After taking action $a_t$ in state $s_t$, the agent receives a reward $R_{t+1} = r(s_t, a_t)$, where $r : S \times A \to \mathbb{R}$. The distribution of initial states is given by $\mu \in \Delta(S)$, and $\gamma \in [0, 1]$ is the discount factor.

An agent may condition its decisions on the interaction history $h_t = (o_0, a_0, o_1, a_1, \ldots, o_t)$. We write $H$ for the set of such histories and define a policy as $\pi : H \to \Delta(A)$, where $\pi(a_t \mid h_t)$ is the probability of taking action $a_t$ after history $h_t$. In our experiments, the policy classes used are either reactive, which condition only on $o_t$, or recurrent, which summarizes the history in a learned hidden state.

A class of problems $(X, F)$ defines a set of POMDPs $\{F(x) : x \in X\}$ generated by a parameter space $X$ and a mapping $F : X \to \mathcal{M}$, where each $x \in X$ encodes a problem's input and $F(x)$ is a POMDP defining $x$. Given a policy class $\Pi$, the goal is to find a $\pi$ that maximizes the return in $F(x)$

$$\arg\max_{\pi \in \Pi} \mathbb{E}_{\pi, p, \mu}[\sum_{k=0}^{\infty} \gamma^k R_{k+1}] . \tag{1}$$

The function $F$ is designed such that a solution to $F(x)$ is a solution to the problem that $x$ defines.

**Example 2.1.** *To illustrate, consider pathfinding problems over a graph $\mathcal{G} = (\mathcal{V}, \mathcal{E})$, where $\mathcal{V}$ is a set of vertices and $\mathcal{E}$ is a set of edges. For an initial and goal vertices $v_0$ and $v_g$ in $\mathcal{V}$, a solution to the problem is a path $P = \{(v_0, v_1), (v_1, v_2), \cdots, (v_k, v_g)\}$, such that each $(v_i, v_j)$ in $P$ is also in $\mathcal{E}$. In this case, $x$ defines the graph, and the initial and goal vertices. The function $F$ defines the pathfinding problem as a POMDP. In this example, the states $S$ are the set of sequences of edges starting from $s_0$. The actions for a state where its sequence of edges ends in $v_k$ are all edges $(v_k, v_i)$ in $\mathcal{E}$ that can be added to the sequence. The transition function $p$ is deterministic, since it returns $1.0$ if the edge added is in $\mathcal{E}$ and $0.0$, otherwise. Since the problem is fully observable, we have that $\Omega = p$. The reward function $r$ returns $0$ once the agent finds the last edge connecting $s_0$ to $s_g$ and $-1$, otherwise. Finally, $\mu$ assigns the value of $1.0$ to $v_0$ and $0.0$ to all other vertices, and $\gamma = 1.0$. With this formulation, a policy that solves Equation 1 can also retrieve a path from $v_0$ to $v_g$ in $\mathcal{G}$.*

**Definition 2.1** (OOD Generalization)**.** Given a class of problems $(X, F)$, a policy $\pi$ generalizes out of distribution if a learner searches in a policy space $\Pi$ for a policy $\pi$ that solves $F(x)$ for all $x$ in $X_{\text{train}} \subset X$, and the resulting $\pi$ also solves $F(x')$ for any $x'$ in $X$.

In practice, $|X_{\text{train}}|$ can be as small as 1, as in the experiments considered in this paper. Moreover, we often cannot prove that the learned $\pi$ solves all $x'$ in $X$. Instead, we sample $x'$ from a set $X_{\text{test}} \subset X$ with $X_{\text{test}} \cap X_{\text{train}} = \emptyset$ to evaluate a policy's capability of generalizing OOD.

**Example 2.2.** *From Example 2.1, if the $\pi$ a learner finds to solve $F(x)$ encodes breadth-first search, such a $\pi$ is guaranteed to solve any $x'$ in $X$.*

The class $\Pi$ determines the biases of the policies we consider. For example, $\Pi$ could be an architecture of a neural network, and the policies $\pi$ within this class are the different weights we can assign to the connections of the neural network. We consider classes $\Pi$ determined by a domain-specific language, so programs written in the language form $\Pi$. A language is defined with a context-free grammar $(\mathcal{N}, \mathcal{T}, \mathcal{R}, \mathcal{I})$, where $\mathcal{N}$, $\mathcal{T}$, $\mathcal{R}$, $\mathcal{I}$ are the sets

of non-terminals, terminals, the production rules, and the grammar's initial symbol, respectively. Figure 1(a) shows an example of a context-free grammar encoding a language for TORCS policies. The grammar's initial symbol $\mathcal{I}$ is $E$. It accepts strings such as the one shown in Figure 1(b), which is obtained through a sequence of production rules applied to the initial symbol: $E \rightarrow$ **if** $B$ **then** $E$ **else** $E \rightarrow$ **if** $B$ **and** $B$ **then** $E$ **else** $E \rightarrow \cdots$.

We compare policy classes given by neural networks and domain-specific languages, which we refer to as programmatic policies, in terms of OOD generalization. We consider TORCS (Verma et al., 2018; 2019), KAREL (Trivedi et al., 2021), and PARKING (Inala et al., 2020) in our experiments.

# 3. Searching for Programmatic Policies

This section describes the algorithms used to synthesize programmatic policies for solving TORCS (Section 3.1), KAREL (Section 3.2), and PARKING (Section 3.3). We aim to provide enough information so the reader understands our results in Section 4. We do not intend to detail the original algorithms. For full method descriptions, see the cited papers in each subsection.

## 3.1. NDPS

Verma et al. (2018) introduced Neurally Directed Program Search (NDPS), a method that uses imitation learning through the DAGGER algorithm (Ross et al., 2011) to learn programmatic policies. Figure 1(a) shows the domain-specific language (Verma et al., 2018) considered in their experiments on the TORCS benchmark. The **peek** function reads the value of a sensor. For example, $\mathbf{peek}(h_{\text{RPM}}, -1)$ reads the latest value (denoted by the parameter $-1$) of the rotation-per-minute sensor ($h_{\text{RPM}}$); $\mathbf{peek}(h_{\text{RPM}}, -2)$ would read the second latest value of the sensor. The $\mathbf{fold}(+, \epsilon - h_i)$ operation adds the difference $\epsilon - h_i$ for a fixed number of steps of the past readings of sensor $h_i$. The non-terminal symbols $P$, $I$, and $D$ in Figure 1(a) form the operations needed to learn PID controllers, with programs that switch between different PID controllers, as shown in Figure 1(b).

NDPS uses a neural policy as an oracle to guide its synthesis. Given a set of state-action pairs $H$, where the actions are given by the neural oracle, NDPS evaluates a program $\rho$ by computing the action agreement of $\rho$ with the actions in $H$. NDPS runs a brute force search algorithm (Albarghouthi et al., 2013; Udupa et al., 2013), to generate a set of candidate programs $C$. Then, it learns the parameters of the programs ($c_1$, $c_2$, and $c_3$ in Figure 1) with Bayesian optimization (Snoek et al., 2012) such that the programs mimic $H$. Once NDPS determines the parameters of programs $C$, it selects the candidate $c$ in $C$ that maximizes the agent's return; $c$ is the starting point of a local search that

optimizes a mixture of the action agreement function and the agent's return. Later, Verma et al. (2019) introduced Imitation-Projected Programmatic Reinforcement Learning (PROPEL), an algorithm that also synthesizes programmatic policies, but it controls how different the oracle can be from the programmatic learner, to ease the imitation learning process. The programmatic policies of both NDPS and PROPEL are called for every state the agent encounters.

## 3.2. LEAPS

Trivedi et al. (2021) introduced Learning Embeddings for Latent Program Synthesis (LEAPS), a system that learns a latent representation of the space of programs a language induces. When given an MDP $\mathcal{M}$, LEAPS searches in the learned latent space for a vector decoded into a program encoding a policy that maximizes the agent's return at $\mathcal{M}$. LEAPS's premise is that searching in the learned latent space is easier than searching in the space of programs, as NDPS and PROPEL do.

Figure 2(a) shows the context-free grammar specifying the language used to encode policies for KAREL. The language accepts programs with conditionals and loops. It also includes a set of perception functions, such as `frontIsClear`, which verifies whether the cell in front of the agent is clear. Further included are action instructions such as `move` and `turnLeft`. The set of perception functions is important because it defines what the agent can observe. As we show in Section 4.2, having access to less information allows the agent to generalize to OOD problems. Figure 2(b) shows an example of a KAREL program. Here, the agent will perform two actions, `pickMarker` and `move`, if a marker is present in its current location; otherwise it will not perform any action.

To learn its latent space, LEAPS generates a data set of programs $P$ by sampling a probabilistic version of the context-free grammar defining the domain-specific language. That is, each production of a non-terminal can be selected with a given probability. A program can be sampled from this probabilistic grammar by starting at the initial symbol and randomly applying production rules until we obtain a program with only terminal symbols. This set of programs is used to train a Variational Auto-Encoder (VAE) (Kingma & Welling, 2014), with its usual reconstruction loss. However, in addition to learn spaces that are more friendly to search algorithms, LEAPS uses two additional losses that attempt to capture the semantics of the programs. These two losses incentivize latent vectors that decode into programs with similar agent behavior to be near each other in the latent space. The intuition is that this behavior locality can render optimization landscapes easier to search.

Once the latent space is trained, it is used to solve MDPs. Given an MDP, LEAPS uses the Cross-Entropy Method

**(a) Domain-Specific Language**

$$
\begin{aligned}
P &::= \mathbf{peek}((\epsilon - h_i), -1) \\
I &::= \mathbf{fold}(+, \epsilon - h_i) \\
D &::= \mathbf{peek}(h_i, -2) - \mathbf{peek}(h_i, -1) \\
C &::= c_1 * P + c_2 * I + c_3 * D \\
B &::= c_0 + c_1 * \mathbf{peek}(h_1, -1) + \ldots \\
  &\quad \cdots + c_k * \mathbf{peek}(h_m, -1) > 0 \mid \\
  &\quad\quad B \textbf{ or } B \mid B \textbf{ and } B \\
E &::= C \mid \textbf{if } B \textbf{ then } E \textbf{ else } E.
\end{aligned}
$$

**(b) Example Policy**

$$
\begin{aligned}
&\textbf{if } (0.001 - \mathbf{peek}(h_{\texttt{TrackPOS}}, -1) > 0) \\
&\quad \textbf{and } (0.001 + \mathbf{peek}(h_{\texttt{TrackPOS}}, -1) > 0) \\
&\quad\quad \textbf{then } 3.97 * \mathbf{peek}((0.44 - h_{\texttt{RPM}}), -1) \\
&\quad\quad\quad + 0.01 * \mathbf{fold}(+, (0.44 - h_{\texttt{RPM}})) \\
&\quad\quad\quad + 48.79 * (\mathbf{peek}(h_{\texttt{RPM}}, -2) - \mathbf{peek}(h_{\texttt{RPM}}, -1)) \\
&\quad\quad \textbf{else } \; 3.97 * \mathbf{peek}((0.40 - h_{\texttt{RPM}}), -1) \\
&\quad\quad\quad + 0.01 * \mathbf{fold}(+, (0.40 - h_{\texttt{RPM}})) \\
&\quad\quad\quad + 48.79 * (\mathbf{peek}(h_{\texttt{RPM}}, -2) - \mathbf{peek}(h_{\texttt{RPM}}, -1))
\end{aligned}
$$

*Figure 1.* (a) Context-free grammar specifying a domain-specific language for TORCS, a racing car domain (Verma et al., 2018). The initial symbol of the language is $E$, $\epsilon$ is a pre-defined constant, and $\{h_i\}_{i=1}^{m}$ is a set of $m$ sensors from which the agent can read. The grammar allows programs that switch between different PID controllers. (b) Example of a policy written in the language.

**(a) Domain-Specific Language**

$$
\begin{aligned}
\rho &:= \textbf{def } \texttt{run m}(s \; \texttt{m}) \\
s &:= \textbf{while } \texttt{c}(b \; \texttt{c}) \; \texttt{w}( \; s \; \texttt{w}) \mid \textbf{if } \; \texttt{c}(b \; \texttt{c}) \; \texttt{i}(s \; \texttt{i}) \mid \\
  &\quad \textbf{ifelse } \texttt{c}(b \; \texttt{c}) \; \texttt{i}(s \; \texttt{i}) \textbf{ else } \texttt{e}(s \; \texttt{e}) \mid \\
  &\quad \textbf{repeat } \texttt{R=}n \; \texttt{r}(s \; \texttt{r}) \mid s; s \mid a \\
b &:= h \mid \textbf{not } (h) \\
n &:= 0, 1, \cdots, 19 \\
h &:= \texttt{frontIsClear} \mid \texttt{leftIsClear} \mid \texttt{rightIsClear} \mid \\
  &\quad \texttt{markersPresent} \mid \texttt{noMarkersPresent} \\
a &:= \texttt{move} \mid \texttt{turnLeft} \mid \texttt{turnRight} \mid \\
  &\quad \texttt{putMarker} \mid \texttt{pickMarker}
\end{aligned}
$$

**(b) Example Policy**

```
def run m(
  if c(markersPresent c) i(
    pickMarker move
  i)
m)
```

*Figure 2.* (a) Context-free grammar specifying a domain-specific language for KAREL. The programs written in this language accept conditional statements and loops. There is a set of perception functions ($h$) and functions that return actions ($a$). (b) Example of a policy for a KAREL task.

(CEM) (Mannor et al., 2003) to search for a vector that decodes into a program that maximizes the return. The rollouts of the decoded policies are used to inform the search.

### 3.3. PSM

Inala et al. (2020) introduced Programmatic State Machine Policies (PSM), a system that learns a policy as a finite-state machine. A finite state machine policy for an MDP $\mathcal{M}$ is a tuple $(M, S, A, \delta, m_0, F, \alpha)$ where $M$ is a finite set of modes. The sets $S$ and $A$ are the sets of states and actions from $\mathcal{M}$. The function $\delta : M \times S \to M$ is the transition function, $m_0$ in $M$ is the initial mode, and $F \subseteq S$ is the set of modes in which the policy terminates. The transition function $\delta$ defines the next mode given the current mode and input state $s$ in $S$. Finally, $\alpha : M \times S \to A$ determines the policy's action when in mode $m$ and the agent observes state $s$.

In the PARKING environment, Inala et al. (2020) considered a domain-specific language for the transition function $\delta$ and constant values for $\alpha$. The grammar defining the language

$\delta$ is the following.

$$
B ::= \{s[i] \geq v\}_{i=1}^{n} \mid \{s[i] \leq v\}_{i=1}^{n} \mid B \wedge B \mid B \vee B
$$

Here, the values $v$ are constants that need to be learned, $s[i]$ is the $i$-th entry of the state $s$ the agent observes at a given time step, and $n$ is the dimensionality of the observation.

## 4. Experiments

In this section, we revisit the experiments of Verma et al. (2018) and of Verma et al. (2019) on TORCS (Section 4.1 and Appendix B), of Trivedi et al. (2021) on KAREL (Section 4.2 and Appendix C), and of Inala et al. (2020) on PARKING (Section 4.3 and Appendix D). In our experiments, we closely follow the methodology used in each of the works we re-evaluated. This includes the choice of learning algorithms, evaluation metrics, presentation of results, and even the dispersion metric used in our tables. This methodological choice explains why the experimental protocols differ across the domains reported in this section.

| Tracks | Ndps Lap Time | Drl ($\beta = 1.0$) Lap Time | Drl ($\beta = 0.5$) Lap Time |
|---|---|---|---|
| G-Track-1 | 1:01 | 54 | 1:17 |
| G-Track-2 (OOD) | 1:40 | CR 1608m | 1:48 (0.76) |
| E-Road (OOD) | 1:51 | CR 1902m | 1:54 (0.69) |
| Aalborg | 2:38 | 1:49 | 2:24 |
| Alpine-2 (OOD) | 3:16 | CR 1688m | 3:13 (1.00) |
| Ruudskogen (OOD) | 3:19 | CR 3232m | 2:46 (1.00) |

*Table 1.* For DRL ($\beta = 0.5$), we trained 30 models (seeds) for G-Track-1 and 15 for Aalborg. Each cell shows the average lap time (mm:ss) over three laps per model, then averaged across models; 13 models learned to complete G-Track-1 and four models learned to complete Aalborg. Values in parentheses for DRL ($\beta = 0.5$) show the fraction of seeds that successfully generalized to the test track (out of 13 and 4 for G-Track-1 and Aalborg, respectively). For Ndps and DRL ($\beta = 1.0$), we used the data from Verma et al. (2018), which is over three models. "CR" indicates that all three models crashed, and the number reported is the average distance at which the agent crashed the car.

### 4.1. Torcs

Verma et al. (2018) and Verma et al. (2019) showed that programmatic policies written in the language from Figure 1 generalize better to OOD problems than neural policies in race tracks of the Open Racing Car Simulator (Torcs) (Wymann et al., 2000). The results of Verma et al. (2018) also showed that neural policies better optimize the agent's return than programmatic policies, as the former complete laps more quickly than the latter on the tracks on which they are trained. We hypothesized that programmatic policies generalize better not because of their representation, but because they are more difficult to optimize than neural policies; as a result, the car moves more slowly, making it easier to generalize to tracks with sharper turns.

We test our hypothesis by training models with two different reward functions: the original function used in previous experiments ($\beta = 1.0$ in Equation 2), which we refer to as "original", and a function that makes the agent more cautious about speeding ($\beta = 0.5$), which we refer to as "cautious".

$$\beta \times V_x \cos(\theta) - |V_x \sin(\theta)| - V_x |d_l| . \qquad (2)$$

Here, $V_x$ is the speed of the car along the longitudinal axis of the car, $\theta$ is the angle between the direction of the car and the direction of the track axis, and $d_l$ is the car's lateral distance from the center of the track. The first term of the reward measures the velocity along the central line of the track, while the second is the velocity moving away from the central line. Maximizing the first term minus the second allows the agent to move fast without deviating from the central line. The last term also contributes to having the agent follow the center of the track. Once we set $\beta = 0.5$, the agent will learn policies where the car moves more slowly, which allows us to test our hypothesis. Equation 2

defines an intrinsic reward, since the evaluation, after the agent is trained, is performed on other metrics: lap time and whether the agent has crashed or not. Therefore, by changing $\beta$ from 1.0 to 0.5 we are not changing the problem, but only how the agent learns to complete a given track.

Following Verma et al. (2018), we use the Deep Deterministic Policy Gradient (DDPG) algorithm (Lillicrap et al., 2019) and Torcs's practice mode, which includes 29 sensors as observation space and the actions of accelerating and steering. We considered two tracks for training the agent: G-Track-1 and Aalborg. The first is considered easier than the second based on the track's number of turns, length, and width. The models trained on G-Track-1 were tested on G-Track-2 and E-Road, while the models trained on Aalborg were tested on Alpine-2 and Ruudskogen.

Table 1 presents the results. Ndps can generalize to the test problems in all three seeds evaluated. Drl with $\beta = 1.0$ does not generalize to the test tracks, with the numbers in the table showing the average distance at which the agent crashes the car in all three seeds. For Drl ($\beta = 0.5$) we trained 30 models (seeds) for G-Track-1 and 15 for Aalborg. Then, we verified that 13 of the 30 models learned how to complete laps of the G-Track-1 track, and 4 of the 15 models learned to complete laps of the Aalborg track; these models were evaluated on the OOD tracks.

The results support our hypothesis that changing the reward function can allow the agent to generalize. On the training tracks, the lap time increases as we reduce $\beta$. Most models trained with $\beta = 0.5$ generalize from the G-Track-1 to G-Track-2 (76% of the models) and E-Road (69%) tracks; all models that learned to complete a lap on Aalborg generalized to the other two tracks.

### 4.2. Karel

Trivedi et al. (2021) showed that programs Leaps synthesized in the language shown in Figure 2(a) generalized better than deep reinforcement learning baselines to problem sizes much larger than those the agent encountered during training. In our experiments, we consider the fully observable version of Karel, where the agent has access to the entire grid, and the partially observable version, where the agent can only perceive the cells around it, as shown by the non-terminal $h$ in Figure 2(a).

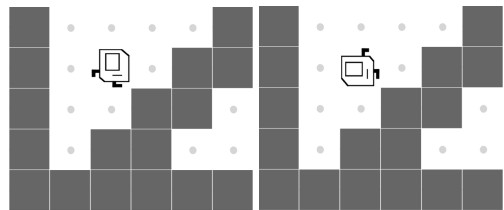

*Figure 3.* Different states but same observation.

|  |  | STAIRCLIMBER | MAZE | TOPOFF | FOURCORNER | HARVESTER |
|---|---|---|---|---|---|---|
| LEAPS[†] | Small | 1.00 (0.00) | 1.00 (0.00) | 0.81 (0.07) | 0.45 (0.40) | 0.45 (0.28) |
|  | 100×100 | 1.00 (0.00) | 1.00 (0.00) | 0.21 (0.03) | 0.45 (0.37) | 0.00 (0.00) |
| PPO with ConvNet[†] | Small | 1.00 (0.00) | 1.00 (0.00) | 0.32 (0.07) | 0.29 (0.05) | 0.90 (0.10) |
|  | 100×100 | 0.00 (0.00) | 0.00 (0.00) | 0.01 (0.01) | 0.00 (0.00) | 0.00 (0.00) |
| PPO with LSTM[†] | Small | 0.13 (0.29) | 1.00 (0.00) | 0.63 (0.23) | 0.36 (0.44) | 0.32 (0.18) |
|  | 100×100 | 0.00 (0.00) | 0.04 (0.05) | 0.15 (0.12) | 0.37 (0.44) | 0.02 (0.01) |
| PPO with $a_{t-1}$ | Small | 1.00 (0.00) | 1.00 (0.00) | 1.00 (0.00) | 1.00 (0.00) | 0.59 (0.05) |
|  | 100×100 | 1.00 (0.00) | 1.00 (0.00) | 1.00 (0.00) | 1.00 (0.00) | 0.04 (0.00) |

*Table 2.* Generalization results on KAREL, where cells show the average return and standard deviation in brackets. "PPO with ConvNet" observes the entire state and employs a convolutional network to learn its representation. "PPO with LSTM" uses an LSTM layer for both actor and critic, while "PPO with $a_{t-1}$" uses a fully connected network with the observation space augmented with the agent's last action. "Small" refers to the problems in which the models were trained, which were of size either $8 \times 8$ or $12 \times 12$. Rows marked with a † are from Trivedi et al. (2021). The results for PPO with $a_{t-1}$ are over 30 seeds, and each seed is evaluated on 10 different initial states; the results for LEAPS and PPO with a ConvNet and with an LSTM are over five seeds and 10 different initial states.

In the partially observable case, the problem cannot, in principle, be solved with fully connected neural networks. Consider the two states shown in Figure 3. In one, the agent is going downstairs; in the other, it is going upstairs. Yet, the observation is the same for both states. Trivedi et al. (2021) used LSTMs (Hochreiter & Schmidhuber, 1997) to deal with partial observability. Instead of using LSTMs, which tend to be more complex to train than fully connected networks, we include the agent's most recent action in the observation. For the fully observable case, we report the results of Trivedi et al. (2021), which used a convolutional network. Trivedi et al.'s results are presented in Table 2 with the labels "PPO with LSTM" and "PPO with ConvNet".

We trained policies for the following problems, which were chosen to match the design of Trivedi et al. (2021) for their OOD generalization experiments: STAIRCLIMBER, MAZE, TOPOFF, FOURCORNER, and HARVESTER. The grid size of these problems was either $8 \times 8$ or $12 \times 12$. After learning to solve these small problems, we evaluated them on grids of size $100 \times 100$, also following Trivedi et al. (2021). In the MAZE problem, the agent learns to escape a small maze and is evaluated on a larger one. Table 2 shows the results.

Our results show that partial observability combined with a simpler model can generalize to larger grids. Namely, "PPO with $a_{t-1}$", which uses a fully connected network with the observation augmented with the agent's last action, generalizes to larger problems. This contrasts with "PPO with ConvNet", which operates in the fully observable setting, and "PPO with LSTM", which operates in the partially observable setting but uses a more complex neural model. To illustrate, in MAZE, if the agent can only see the cells around itself, it can learn strategies such as "follow the right wall", which is challenging to learn in the fully observable setting. The LSTM agent fails not only to generalize to larger problems, but it often also fails to learn how to solve

even the smaller problems.

### 4.3. PARKING

In PARKING, the agent must exit a parking spot. During training, the gap between cars is sampled uniformly from $[12.0, 13.5]$, while the test range is $[11.0, 12.0]$, requiring generalization to tighter scenarios. We evaluate both programmatic policies, as described by Inala et al. (2020), and neural policies trained using DQN (Mnih et al., 2015). Preliminary experiments showed that DQN outperformed the PPO and DDPG algorithms considered in our previous experiments. For each policy type, we trained 30 independently seeded models and evaluated each one on 100 test episodes, where the test gap was sampled uniformly from the range $[11.0, 12.0]$.

Table 3 shows the results. We trained 30 independent models of PSM and 15 of DQN. Each model was evaluated on 100 different initial states. The columns "Successful-on-100" refer to the ratio of models that could solve all 100 initial states. For example, $0.06$ for PSM means that two of the 30 models solved all initial states on training and test. The "Successful Rate" column shows the ratio of times across all models and initial states that the learned policy could solve the problem. For example, $0.86$ for DQN in training means that DQN models solved 86% of the $15 \times 100 = 1500$ initial states.

Our results suggest that the PSM policies generalize better than the DQN policies, as two out of 30 models could solve all 100 test initial states. Looking at the difference between the "Success Rate" of PSM and DQN in training and test also suggests that PSM's policies generalize better, as the gap between the two scenarios is small for PSM: $0.26 - 0.16 = 0.10$ versus $0.86 - 0.18 = 0.68$ for DQN. However, looking at the test "Success Rate" alone suggests that DQN is the

| | PSM | | DQN | |
|---|---|---|---|---|
| | Successful-on-100 | Success Rate | Successful-on-100 | Success Rate |
| Training | 0.06 (0.09) | 0.26 (0.13) | 0.40 (0.22) | 0.86 (0.14) |
| Test | 0.06 (0.09) | 0.16 (0.12) | 0.00 (0.10) | 0.18 (0.08) |

*Table 3.* Evaluation of 30 seeds of PSM and 15 seeds of DQN on the PARKING domain. Each model trained was evaluated on 100 different initial states of both training and testing settings. The columns "Successful-on-100" report the fraction of models trained that successfully solved all 100 initial states. The columns "Success Rate" report the average number of initial states solved across different seeds. We also present the 95% confidence intervals in brackets.

winner, as DQN policies can solve more test initial states on average than PSM can. Independent of the metric considered, our results show that PARKING is a challenging domain for both types of representation.

### 4.4. Discussion

Our experiments showed that neural models can also generalize to OOD problems commonly used in the literature. One key aspect of programmatic solutions is the policy's sparsity. For example, the mode transitions in Figure 5 use a single variable in the Boolean expression. By contrast, neural networks typically use all variables available while defining such transitions, often by encountering spurious correlations between input features and the agent's action. That is why providing fewer input features, combined with a simpler neural model, helped with generalization in KAREL—we remove features that could generate spurious correlations with the model's actions. These results on reducing input features to enhance generalization align with other studies involving the removal of visual distractions that could hamper generalization (Bertoin et al., 2022; Grooten et al., 2024).

In the case of TORCS, OOD generalization was possible due to a safer intrinsic reward. If the agent learns on a track that allows it to move fast and never slow down, then it is unlikely to generalize to race tracks with sharp turns that require the agent to slow down. In this case, generalization or lack thereof is not caused by the representation, but by how well the agent can optimize its return while using that representation. We conjecture that NDPS and PROPEL would not generalize well if they could find better optimized policies for the agent's return in the training tracks.

PARKING was the most challenging benchmark we considered in our experiments, and we believe it points in the direction of benchmarks that could distinguish the generalization power of programmatic and neural representations. Recurrent neural networks can, in principle, represent the solution shown in Figure 5. In fact, due to the loop of the agent interacting with the environment, the solution to PARKING does not even require loops. If we augment the agent's observation with its last action, a decision tree could encode

the repetitive behavior needed to solve the problem. Yet, we could not find either a neural policy or a programmatic one that reliably generalizes to OOD problems in PARKING.

The domain-specific languages used in our three domains induce spaces similar to those of neural networks. For example, TORCS's language (Figure 1) allows programs with if-then-else chains and trainable parameters $c_0, \ldots, c_k$, a space resembling that of ReLU networks (Orfanos & Lelis, 2023). The ReLU space can be made a superset of the TORCS language by providing the **peek** and **fold** functions, as shown in Figure 1, as network inputs and varying the number of neurons so the network programs can also grow in length—larger networks represent longer programs. In this case, both programmatic and neural spaces are expressive, as they contain solutions that generalize, and differences in OOD generalization can be attributed to discoverability.

## 5. When Should We Use Programs?

For which type of problem do commonly used neural architectures not satisfy either the expressivity or discoverability property? As suggested by our experiments and those in the literature, controlling for the discoverability property can be challenging because it depends on search heuristics that may not be initially obvious. We offer illustrative answers to this question by controlling for the expressivity property.

In KAREL's Maze, shown in Figure 4, the agent needs to find a path from its initial location to the marker (light-gray object), while sensing only the cells adjacent to the agent. Since all corridors of the maze are one-cell wide, wall-following strategies can solve this problem. The neural models considered in our experiments can encode a wall-following algorithm because such algorithms have a constant-memory requirement. Given the direction the agent is moving, which can be given by the agent's last action, and the wall sensors, the next action is computable in $O(1)$ memory per step. Since the width of feedforward models and the hidden state of recurrent models are fixed and independent of the input, they cannot encode algorithms whose working memory grows with the input size.

To illustrate, consider pathfinding over graphs $\mathcal{G} = (\mathcal{V}, \mathcal{E})$

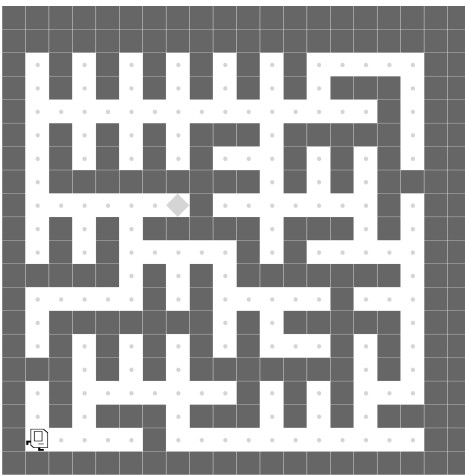

*Figure 4.* KAREL's Maze.

(Example 2.1), of which KAREL's Maze is a special case. The memory requirement of exact algorithms grows with input size: breadth-first search maintains a frontier and visited set of size $\Theta(|\mathcal{V}|)$; iterative-deepening depth-first search has a memory requirement of $\Theta(d)$, where $d$ is the solution depth, which can grow with the input size. Independent of any specific algorithm, simply indexing a vertex among $|\mathcal{V}|$ candidates requires $\Omega(\log |\mathcal{V}|)$ bits, so a constant-memory representation is insufficient. The neural policies we evaluated have fixed-sized hidden states, which are independent of $|\mathcal{V}|$. These fixed-capacity policies cannot represent general pathfinding algorithms whose working memory grows with the input size, and thus should not be expected to generalize to pathfinding instances that require such memory.

In addition to pathfinding, constant-memory models cannot guarantee OOD generalization in benchmarks that exhibit nested subproblems, such as NetHack (Hambro et al., 2022). In these problems, the agent is interrupted to solve an inner task and must later resume the outer one. Correct handling of arbitrarily deep nesting requires maintaining a stack of pending contexts, used to push a subproblem on interruption and pop it on completion. When the nesting depth can grow with the instance, fixed-capacity models cannot guarantee correctness or OOD generalization, as they cannot represent a stack that grows depending on the input problem. In addition to the inability of these neural models to grow their memory capacity to match the needs of the input, they have been shown to fail to learn stack-like structures (Joulin & Mikolov, 2015). This suggests that such models may fail even on problems where the required memory fits within the model's fixed capacity, because the relevant stack-like computation may still be difficult to discover from data.

In contrast to the neural models considered, whose memory capacity is constant and determined at training time, programmatic representations can produce policies whose memory capacity grows according to the input size. An algorithm whose working memory usage is a function of the input size can generalize to larger instances, whereas a fixed-capacity model cannot. Although recurrent models are, in theory, computationally universal (Siegelmann & Sontag, 1994; 1995), recent work has shown that they are more limited, theoretically (Nowak et al., 2023) and empirically (Delétang et al., 2023). For example, even when the model has the memory capacity required to solve the problem, increasing the memory needed can lead to imprecise computation, as Weiss et al. (2018) showed in their counting experiment: while LSTMs can learn to count so they can recognize languages such as $a^n b^n$, as the value of $n$ grows during test time, the model starts to present what they called a "slightly-imprecise counting" behavior and fail to generalize to large $n$. By contrast, a programmatic representation could implement a pushdown automaton that provably generalizes for any value of $n$.

One might be tempted to approach the OOD problem by scaling the network size: if the model needs to handle larger inputs, one could train a larger model with more capacity. This does not solve the OOD generalization problem because one can always provide larger inputs that the model cannot handle. This issue is analogous to the relationship between finite-state machines (FSMs) and pushdown automata (PDAs): for any fixed bound on input size, one can construct a sufficiently large FSM to handle that bound, but this does not imply that FSMs are as expressive as PDAs. For problems whose solutions require growing memory, only a representation with the appropriate computational mechanism can generalize to arbitrary input sizes.

Neural architectures inspired by formal-language methods, including memory-augmented models such as stack-RNNs (Joulin & Mikolov, 2015) and neural Turing machines (Graves et al., 2014), point to a promising research direction. Because they attempt to bring together the best of both worlds, they shift the question from whether a policy should be represented as a program or a neural network to whether the chosen representation has the computational structure needed to solve the underlying problem. In our terminology, these works attempt to design representations that are expressive by satisfying the computational requirements of the problem, while also attempting to make generalizing solutions discoverable through gradient-based optimization.

## 6. Relation to Other Works

Our findings may have implications for other studies comparing the generalization of programmatic and neural policies. For example, (Cui et al., 2024) evaluated their method on Karel tasks using a recurrent neural policy baseline; our experiments show that a simpler feedforward network augmented with the agent's last action can improve discov-

erability in this setting. (Guo et al., 2023) represent policies as symbolic equations. Since standard neural networks can approximate these functions arbitrarily well, the two representations are equivalent in expressivity, and any observed generalization gap could be the result of differences in discoverability. (Qiu & Zhu, 2022) also reported generalization advantages of programs represented by a chain of if-then-else structures over a recurrent model. The programs make calls to predefined functions that encode agent behaviors, such as a move-left function. Feedforward networks could represent the same space of programs if their outputs defined a mixture of the values returned by the functions, as options are often used in RL (Sutton et al., 1999; Bacon et al., 2017). Although a careful investigation is needed, Qiu & Zhu's reported advantage of programmatic representations may also be attributed to discoverability confounding factors.

Work in robotics has investigated generalization in contexts different from the distribution shift we consider in our work. For example, Holtz et al. (2021a;b) considered generalization in the context of policy repair, while Xin et al. (2024) considered generalization in the context of robustness to noise. These works compare programmatic representations to neural or learning-based alternatives along dimensions such as sample efficiency and robustness. However, in contrast to the papers we re-evaluated, they do not directly compare fixed neural and programmatic policies under the same OOD distribution-shift setting considered in our work.

Other works adopt hybrid representations that combine the perceptual strengths of neural networks with the algorithmic structure of programmatic components (Qiu et al., 2023), which have the potential to overcome the memory limitations of standard neural architectures while still benefiting from neural models' ability to handle perception tasks, as in the Houdini system (Valkov et al., 2018).

Although we focused on generalization, programmatic representations offer additional benefits, such as interpretability (Kohler et al., 2024) and improved sample efficiency (Qiu et al., 2023). Some of the benefits related to sample efficiency stem from the modular nature of programs, which enables the reuse of subprograms (Liu et al., 2023). In this case, the subprograms need to generalize to enable reuse. Although the reuse of subprograms is more common with programmatic representations, previous work has also investigated the reuse of subprograms in neural representations through the decomposition of networks (Alikhasi & Lelis, 2024) and policy composition (Qureshi et al., 2020).

## 7. Conclusion

In this paper, we re-evaluated prior claims that programmatic representations generalize better than neural policies in reinforcement learning. By revisiting representative experiments from the literature, we showed that much of the reported generalization gap can be attributed to uncontrolled experimental factors rather than to inherent representational differences. When training pipelines are carefully controlled, neural policies can achieve levels of OOD generalization comparable to those reported for programmatic policies in the same domains. To interpret these findings, we distinguished between two properties of a policy representation: expressivity, the ability to encode a solution that generalizes, and discoverability, the ability of a given search or optimization procedure to recover such a solution in practice. Our results suggested that existing experiments from the literature for evaluating OOD generalization do not separate the expressivity effects from the discoverability effects of a given representation. Clarifying this distinction is necessary to understand comparisons of OOD generalization across different representations. More broadly, our results suggest that the discussion of representation learning in the context of OOD generalization should shift from the choice between programmatic and neural approaches to the design of representations that meet the computational requirements of the underlying problem. If a problem requires memory or other forms of computation that grow with the input size, then the representation should be expressive enough to encode solutions with those properties. At the same time, expressivity alone is not sufficient: the learning algorithm must also be able to discover such solutions in practice. We believe that focusing on both expressivity and discoverability can help guide the design of future representations that enable OOD generalization.

## Acknowledgements

This research was supported by Canada's NSERC and the CIFAR AI Chairs program. This research was enabled in part by support provided by the Digital Research Alliance of Canada. The authors thank the anonymous reviewers for valuable feedback on this work.

## Impact Statement

This paper presents work whose goal is to advance the field of machine learning by revisiting and re-evaluating claims made in previous work. There are many potential societal consequences of our work, none of which we feel must be specifically highlighted here.

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

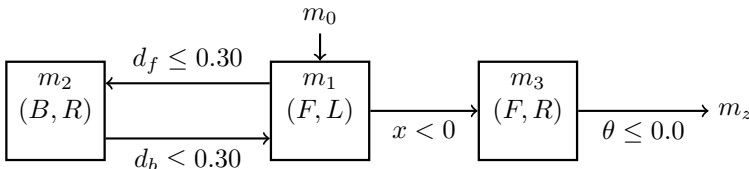

*Figure 5.* Example of a state machine policy, where $m_0$ is the initial mode and $m_z$ is an accepting mode. The tuples inside each mode specify the agent's action when in that mode (e.g., $(F, L)$ means "move forward and steer to the left". The transitions from one mode to another are triggered by a Boolean expression shown in the arrows. For example, if the car is too close to the car in front of it ($d_f \leq 0.30$), then the policy moves from $m_1$ to $m_2$. The agent remains in the current mode if no outgoing Boolean expression is triggered. This policy is based on an example by Inala et al. (2020).

## A. Example PSM

Figure 5 shows an example of the type of policy PSM learns. In this example, the policy is for PARKING, a domain where the agent must learn how to exit a parking spot with a car in front of the agent's car ($car_f$) and another at the rear ($car_b$). The policy uses the following state features: the distance between the agent's car and $car_f$ ($d_f$) and $car_b$ ($d_b$), the $x$ coordinate of the car, and the angle $\theta$ of the car. A solution involves the agent moving forward to the left (mode $m_1$) and then back to the right (mode $m_2$), until the agent has cleared $car_f$ (transitioning to mode $m_3$). The agent solves the problem if it straightens the car after clearing $car_f$, thus transitioning from $m_3$ to $m_f$. PSM's policies are called only once for the initial state; the policy returns only at the end of the episode.

## B. TORCS Details

We use the hyperparameters in Table 4 with DDPG (Lillicrap et al., 2019).

| Hyperparameter | Selected Value |
|---|---|
| Actor's learning rate | 0.0003 |
| Critic's learning rate | 0.001 |
| Batch size | 64 |
| Buffer size | 100000 |
| $\tau$ | 0.005 |
| L1 regularization | 0.00001 |
| Max steps | 20000 |
| Training episodes | 600 |

*Table 4.* Hyperparameter Configuration Used for TORCS

## C. KAREL Details

We used Proximal Policy Optimization (PPO) (Schulman et al., 2017) with the agent's previous action appended to the observation vector. A comprehensive hyperparameter sweep was conducted over the values from Table 5.

The max steps used for training and testing different Karel tasks are shown in Table 6.

Table 7 shows the best-performing configuration across all five tasks for final evaluation. The selection was based on the agent's average return across the three seeds after two million time steps of training.

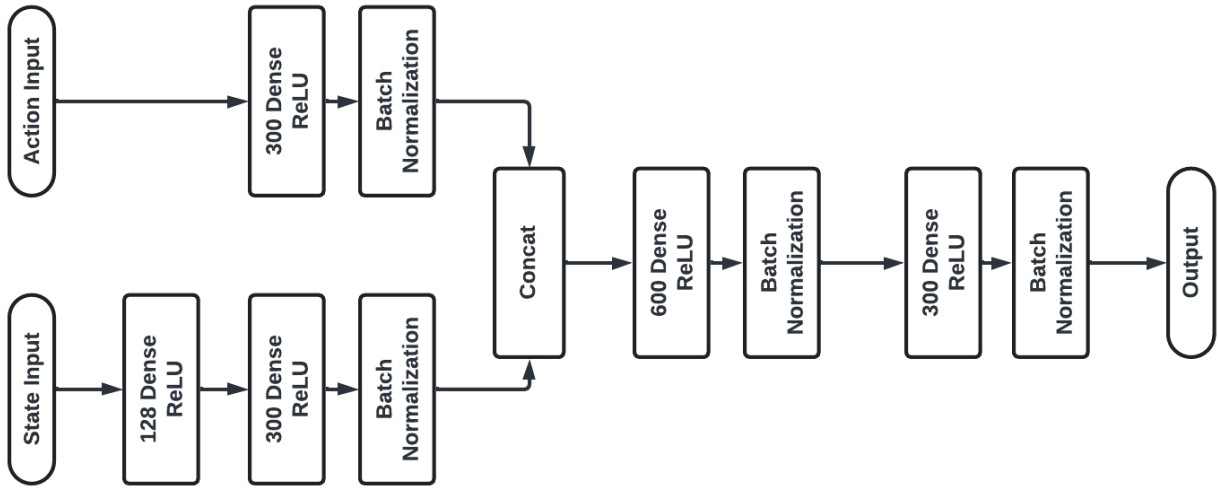

*Figure 6.* Architecture of the critic network used in DDPG for the TORCS environment.

| Hyperparameter | Values Tested |
|---|---|
| Learning rate | $\{0.001, 0.0001, 0.00001\}$ |
| Clipping coefficient | $\{0.01, 0.1, 0.2\}$ |
| Entropy coefficient | $\{0.001, 0.01, 0.1\}$ |
| L1 regularization | $\{0.0, 0.0001, 0.0005, 0.001\}$ |
| Actor's hidden layer size | $\{32, 64\}$ |
| Training time steps | $\{2\ million\}$ |
| Seeds per config | $\{3\}$ |

*Table 5.* PPO + Last Action: Hyperparameter Sweep Configuration for Karel

| Hyperparameter | Selected Value |
|---|---|
| Learning rate | 0.001 |
| Clipping coefficient | 0.1 |
| Entropy coefficient | 0.1 |
| L1 regularization | 0.0 |
| Actor's hidden layer size | 32 |

*Table 7.* Best Hyperparameter Configuration Used for Final Training of Karel

The best configuration was then trained with 30 random seeds, and evaluation results were averaged over 10 distinct initial configurations per seed. Additionally, four other hyperparameter configurations achieved 100% generalization on four out of five tasks: STAIRCLIMBER, MAZE, and TOPOFF.

Training used diverse initial state configurations. Whenever feasible, we enumerated all combinations of agent and goal placements. Specifically:

- For STAIRCLIMBER, TOPOFF, and FOURCORNER, all possible agent-goal placements were used.

- For MAZE, where full enumeration was computationally infeasible, we sampled 5 random mazes and placed the goal at every position on the grid.

|  | STAIRCLIMBER | MAZE | TOPOFF | FOURCORNER | HARVESTER |
|---|---|---|---|---|---|
| TRAINING | 50 | 100 | 100 | 100 | 200 |
| TEST | 1000 | 100000 | 1000 | 1000 | 10000 |

*Table 6.* Max steps of episodes for each Karel task during training and test.

The training grid sizes for each task were:

- $12 \times 12$ for STAIRCLIMBER, TOPOFF, and FOURCORNER

- $8 \times 8$ for MAZE and HARVESTER

## D. PARKING Details

We used a single-hidden-layer DQN architecture with 64 units for the neural baseline. The agent operated over a discretized action space, where continuous actions were mapped onto $n$ equally spaced values using a fixed action resolution. We performed a grid search over the hyperparameter values listed in Table 8. The selected hyper-parameters are shown in Table 9.

| Hyperparameter | Values Tested |
|---|---|
| Learning rate | $\{0.01, 0.001, 0.0001\}$ |
| Batch size | $\{64, 128, 256\}$ |
| Target update frequency | $\{100, 500, 1000\}$ |
| $\epsilon$ | $\{0.1, 0.01\}$ |
| Replay buffer size | $\{1\ \text{million}, 2\ \text{million}\}$ |
| Action resolution | $\{3, 5, 7\}$ |
| Seeds per config | $\{10\}$ |

*Table 8.* DQN: Hyperparameter Sweep Configuration for Parking Domain

| Hyperparameter | Selected Value |
|---|---|
| Learning rate | $0.0001$ |
| Batch size | $64$ |
| Target update frequency | $1000$ |
| $\epsilon$ | $0.01$ |
| Action resolution | $3$ |

*Table 9.* Best Hyperparameter Configuration Used for Final Training

The original PARKING benchmark introduced by Inala et al. (2020) was not designed with reinforcement learning in mind—it provides "safety check" to invalidate policies that crash the car or get out of boundaries. We define both a shaped reward function and a termination condition to adapt it for RL. If the agent successfully reaches the parking exit, the episode ends with a large positive reward ($2 \times$ max episode length); if it takes an unsafe action, it terminates immediately with a large negative penalty ($-2 \times$ max episode length). Otherwise, at each timestep the agent receives $r_t = -\left(2|x_{\text{agent}} - x_{\text{goal}}| + |y_{\text{agent}} - y_{\text{goal}}|\right) - 1$, i.e., the (weighted) negative Manhattan distance minus an extra step penalty of 1, encouraging the car to move closer to the exit.

