# OpenReview forum: "Revisiting OOD Generalization in Programmatic RL"
_ICML.cc/2026/Conference — ICML 2026 regular_

### Official Review · Reviewer_oMuK · 2026-03-05

**Soundness:** 3
**Presentation:** 3
**Significance:** 2
**Originality:** 2
**Overall Recommendation:** 4
**Confidence:** 1

**Summary:**

This paper revisits prior claims that programmatic policies generalize better than neural policies in reinforcement learning tasks. The authors argue that previously reported performance gaps arise largely from experimental confounds rather than intrinsic representational advantages.

By introducing several modifications to the neural training pipeline, the authors show that neural policies can match or even outperform programmatic policies in out-of-distribution (OOD) generalization.

**Compliance With Llm Reviewing Policy:**

Affirmed.

**Key Questions For Authors:**

N/A

**Limitations:**

No, this paper did not discuss limitations or potential societal impact.

**Strengths And Weaknesses:**

Strengths:

1. The paper addresses a central question in RL: whether structured programmatic policies inherently generalize better than neural networks. This is important because many recent works implicitly rely on this assumption.

2. The experiments suggest that neural policies may fail to generalize because of overly rich observations and eward structures, which are is interesting and aligns with known issues in RL generalization. The paper identifies several concrete issues in prior setups, such as suboptimal solutions and unintended objectives. These are well-known issues in RL, but the paper connects them directly to OOD generalization gaps, which is valuable.

Weakness:

1. The paper mainly performs experimental re-evaluation rather than introducing a new algorithm. The paper mainly relies on modifying observation space, adjusting reward functions or tuning training procedures. Most modifications to neural training appear relatively straightforward, which may limit the methodological novelty.

2. Although the paper introduces the expressivity vs discoverability framework, the analysis remains mostly conceptual. For example, there is no formal comparison of policy space expressivity. A deeper theoretical or empirical analysis would strengthen the argument.

---

> ### Author Rebuttal · Authors · 2026-03-26
>
> Thank you for your feedback.
>
> **1 - Limited Methodological Novelty**
>
> Our paper is unconventional, and as such, its originality shouldn't be judged as a paper that introduces a novel methodology.  Also, using existing methods to contest existing claims makes our contribution much cleaner. We are proud to not introduce anything new to show that previous claims are incorrect. Our novelty lies in understanding existing results. In particular, we now understand that programmatic representations shouldn't be our default choice for OOD generalization, and why that is the case. We show that the choice of representation is much more nuanced and depends on the computational requirements of the underlying problem.
>
> **2 - Formal Comparison of Expressivity**
>
> We considered this, but it is already well explained in previous work. Instead, we relied on existing work to provide a formal comparison in terms of expressivity. Please see the paragraph above Section 5.

---

### Official Review · Reviewer_6sd6 · 2026-03-10

**Soundness:** 2
**Presentation:** 2
**Significance:** 2
**Originality:** 1
**Overall Recommendation:** 4
**Confidence:** 4

**Summary:**

The work studies the generalization capabilities of programmatic policies with neural policies learned through reinforcement learning (RL). Experiments are conducted in three benchmark environments, Torcs, Karel und the Parking environment, and compare RL-learned policies with the programmatic policy algorithms Neural Directed Program Search (NDPS), Learning Embeddings for Latent Program Synthesis (LEAPS), and Programmatic State Machine Policies (PSM). Experiments in Torcs and Karel show that out-of-the-box RL learned policies generalize worse to new held-out environments compared to programmatic policies obtained by NDPS and LEAPS. However, through reward-shaping or providing additional input features in Torcs and Karel, respectively, policies learned through RL match or exceed the performance of these programmatic approaches in held-out tasks. Generalization in the Parking environment is found to be challenging for both RL and programmatic approaches. The work explains these findings to the expressivity and discoverability of the policy representation.

**Compliance With Llm Reviewing Policy:**

Affirmed.

**Final Justification:**

The authors' rebuttal addressed many of my concerns. My minor concerns have all been addressed. My major concerns regarding the significance and originality have been partially addressed. In particular, I believe that discussion with the authors gave me a clearer picture of the contributions of the work and how these findings and perspectives could be significant contributions to advance the field. I continue to consider the work borderline but have revised my score to reflect that I am more leaning towards acceptance now.

**Key Questions For Authors:**

1. For the experiment in Karel, Table 2 shows results of LEAPS. The 2nd paragraph in 4.2 mentions both a variant of LEAPS using CNNs for the fully-observable task as well as a variant using LSTMs for the partially-observable task. Which of these are you evaluating in Table 2?
2. Why did you decide to use different RL algorithms and architectures across all three tasks (Torcs using DDPG, Karel using PPO, and Parking using DQN)?

**Limitations:**

Yes.

**Strengths And Weaknesses:**

## Strengths
The motivation and premise of this work are highly relevant and significant to the RL community. Programmatic/ symbolic policy representations are often studied separately from neural policy representations so gaining a richer understanding of which approach is most suitable under which conditions is highly valuable. In particular Section 5 seemed very relevant in that regard. Unfortunately, I believe that the depth and breadth of this work and its study towards these questions is lacking, as outlined below, such that I cannot recommend acceptance.

## Weaknesses
Below, I provide a list of weaknesses of this work. Any major concern would need to be thoroughly addressed to warrant acceptance.

### W1: Lacking Significance (Major)
I believe that this work is mostly lacking in significant contributions.

In its current form, it is unclear to me what general findings to take from this work that would motivate/ enable follow-up work. To enable RL policies to generalize better, the authors use reward-shaping and feature engineering in Torcs and Karel, respectively. In the conclusion, these modifications are framed as carefully controlling experimental factors but I would rather see them as task-specific modifications to facilitate RL generalization. These are reasonable approaches to get the most out of RL algorithms, and indicates the potential of RL policies under the right conditions but from these findings I mainly read "with some work on feature and reward engineering, RL can learn generalizable policies" without adding further nuance to already existing work that discusses such techniques.

Towards the end of Section 5, the authors state that their findings "point to a promising research direction that combines the strengths of neural and programmatic representations" but to the best of my knowledge this is already pursued and even done in several of the studied programmatic policy algorithms in this work. NDP leverages neural policies to guide its synthesis process and LEAPS learns a latent space using a neural network approach to facilitate efficient search. Again, I struggle to extract new methodology or findings that would constitute significant contributions.

### W2: Originality (Major)
The work is limited in its originality. It evaluates existing programmatic policy approaches with standard RL algorithms and deploys known techniques to improve the ability of RL policies (reward-shaping and feature engineering) in existing benchmarks. I want to be clear that these limitations are per se no reason for rejection if these standard techniques enable novel and significant findings that, for example, point towards new promising research directions or problems. However, as outlined above in W1, I believe that the work falls short in this regard.

### W3: Soundness - Inconsistencies across Experiments
It is partially difficult to extract general findings from this work because the three conducted experiments in Torcs, Karel and Parking differ in key properties, including the setup of training and testing tasks and RL algorithms. I would like to emphasize that these deviations are not necessarily major concerns but they should be deliberate choices and carefully justified. These justifications are currently absent.

The RL algorithm changes from environment to environment. In Torcs, policies are trained with DDPG, in Karel with PPO, and in Parking with DQN. This is not per se a problem since it also expands the set of tried algorithms, but these choices are not justified or tied to any particular properties or goals. Instead, they appear like non-deliberate deviations that make it harder to identify under which conditions these findings are expected.

Similarly, for a study focused on OOD generalization, I would expect a deliberate choice of how to setup and separate training and testing tasks. However, these details vary across all three experiments and, again, lack any clear motivation or justification for these changes. In Torcs, agents appear to be trained in a single task and evaluated in completely new tasks/ maps. In Karel, agents are trained in two grid sizes of a particular task and are then evaluated in the same task but with significantly larger grid sizes (from 8x8 and 12x12 during training to 100x100 during testing). In Parking, training is done on a randomized range of parking gaps of [12.0, 13.5] while testing is done on a slightly deviating smaller range of [11.0, 12.0] to constitute harder tasks.

### W4: Soundness/ Presentation - Unclear Setup for LEAPS in Karel
From the 2nd paragraph in Section 4.2, it appears that LEAPS was originally proposed with two variants for the Karel benchmark: one working on the fully-observable Karel tasks with CNN networks and one for the partially observable tasks with LSTM networks. Table 2 presents results of the LEAPS algorithm in the Karel benchmark -- which variant of LEAPS are you using to obtain these results? Generally, it was not clear to me at first that the different algorithms in Table 2 are evaluated in different versions of these tasks; I'd highly suggest to clarify this more explicitly.

### W5: Soundness - Misleading Statement
In the 3rd paragraph of Section 5, the work states "The neural policies we evaluated have fixed-size hidden states, which are independent of $|\mathcal{V}|$. Consequently, they are not expressive to represent instance-growing structures and thus cannot generalize OOD in pathfinding problems." I find these statements slightly misleading since they seem to imply that neural policies are ill-suited for problems that grow in complexity. While it is true that a small neural network might lack capacity to represent a policy capable of solving a larger problem, it is perfectly feasible to train a network of larger capacity that could solve the required problems. Also, as acknowledged later, recurrent networks for instance can express problems that require growing memory so this statement should likely be made more precise as to which setup/ policies are being assumed.

### Minor Comments
1. The definition of POMDPs in Section 2 differs from standard definitions in two places. (1) The reward function is defined on the observations and actions rather than the ground-truth state and action, and (2) the policy is stated to be conditioned on the current observation while under the POMDP formalism, the policy should typically be conditioned on the full observation history as the most detailed approximation of the environment state.

---

> ### Author Rebuttal · Authors · 2026-03-26
>
> Thank you for your feedback.
>
> **1 - Lack of Significance**
>
> We agree that tricks to make neural policies generalize are not interesting contributions, and we don't claim them as such. The use of these tricks was to show that the claims from the literature were incorrect.
>
> The significance of the paper is to demonstrate that we shouldn't assume programmatic representations are automatically superior to neural ones for OOD generalization. This is the message the papers we studied imply. Instead, we should understand the computational requirements to decide whether we need a programmatic or a neural representation for a given problem.
>
> If the programs we can write are as expressive as the neural networks we can train, then it is advisable to consider using neural networks, as they tend to be much easier to train and do not require special combinatorial search procedures or the time-consuming design of DSLs.
>
> If the problem requires a specific type of computation to achieve OOD generalization that neural networks cannot provide, then we should consider programmatic representations that can meet those requirements.
>
> This guidance is missing in the literature, and our paper fills this gap.
>
> By combining neural and symbolic approaches, we aim to draw inspiration from programmatic representations to design novel neural architectures that can meet the computational requirements that current models cannot. One example is the Stack-Augmented RNNs from the NLP community. We believe we need to investigate similar lines of work in RL.
>
> Our paper also points to the types of problems the community might want to consider when comparing programmatic and neural representations, such as the nested subproblems found in NetHack.
>
> Overall, our work provides several directions for future research: guidelines for selecting a suitable representation, novel neural architectures that enable OOD generalization, and the design of benchmarks to evaluate OOD generalization.
>
> We will be happy to clarify these points in the paper.
>
> **2 - Originality**
>
> Our paper is unconventional, and as such, its originality shouldn't be judged as a paper that introduces a novel algorithm. Our paper is intended to revisit existing experiments in the literature, so we cannot claim originality in that regard. Also, using existing methods to contest existing claims makes our contribution much cleaner. We are proud to not introduce anything new to show that previous claims are incorrect. Our originality lies in understanding existing results. In particular, we now understand that programmatic representations shouldn't be our default choice for OOD generalization, and why that is the case. We show that the choice of representation is much more nuanced and depends on the computational requirements of the underlying problem.
>
> **3 - Inconsistencies across Experiments**
>
> We are revisiting experiments from three papers by three different research groups. It is expected that the experiments will follow different methodologies. In fact, it would be very strange if they didn't. All settings we used exactly mimic those used in the papers we studied, including the choice of learning algorithm, training/test split, and hyperparameters. We wouldn't be able to claim that we are revisiting previous experiments if we didn't do it that way.
>
> **4 - Unclear Setup for LEAPS in Karel**
>
> We present all variants in our Table 2. LEAPS uses only the partially observable space. It is the neural baselines from the LEAPS paper that consider both the entire state (CNN) and the partially observable state (LSTM); they are both presented in our table. We will clarify that in the paper.
>
> **5 - Misleading Statement**
>
> The idea that we can always train a larger neural network to attain OOD generalization is a common misconception that our paper addresses.
>
> In OOD generalization, we want to handle inputs that weren't seen during training. After training, if we are given an input larger than those seen during training, we might decide to retrain a larger model to handle it. However, we can always be given slightly larger inputs, thus triggering the training of an even larger model. This is clearly not a solution to OOD generalization (see Definition 2.1). A program that can handle growing input size is a solution to OOD generalization: we train it once, and it generalizes to all inputs.
>
> This misconception is equivalent to the one that finite state machines (FSMs) are equivalent to pushdown automata (PDA) because we can always create a larger FSM to handle larger inputs. The problem is that we can always provide even larger inputs that would break the FSM (no OOD generalization). That is why PDAs are more expressive than FSMs, like the programs and neural models with fixed capacity discussed in our paper.
>
> So, our statement isn't misleading, but we can improve it with an explanation similar to the one above.
>
> **6 - Key Questions for Authors**
>
> Please see points 3 and 4 above.

---

> > ### Author Rebuttal · Reviewer_6sd6 · 2026-04-01
> >
> > I thank the authors for their clarifications which addressed some misunderstandings of my original review.
> >
> > **1 - Significance:** I thank the authors for elaborating on their view. I agree that clear guidance or suggestions for open research questions are significant contributions in their own right. However, I remain doubtful to what extent this work achieves that goal. The authors state
> >
> > > If the programs we can write are as expressive as the neural networks we can train, then it is advisable to consider using neural networks, as they tend to be much easier to train and do not require special combinatorial search procedures or the time-consuming design of DSLs. If the problem requires a specific type of computation to achieve OOD generalization that neural networks cannot provide, then we should consider programmatic representations that can meet those requirements. This guidance is missing in the literature, and our paper fills this gap.
> >
> > but one could equally read from the results in Torcs and Karel that neural networks can be beneficial over often more complex programmatic approaches even when specific types of computation are required. If one knows which computation is required, through feature engineering or reward shaping based on this domain knowledge, neural policies are shown to perform just as well or better than programmatic policies. In that sense, I would not agree with the author's summary of the guidance read from their work. Maybe the summary should be that domain knowledge through informed design of programmatic representations or through reward/ feature engineering for neural approaches can boost generalization, independent of whether programmatic or neural policies are being deployed.
> >
> > **2 - Originality:** I agree that each paper should be evaluated on its own. I do not expect the work to propose a novel algorithm nor have I claimed such is necessary to show originality.
> >
> > > We show that the choice of representation is much more nuanced and depends on the computational requirements of the underlying problem.
> >
> > What do you mean with the computational requirements of the underlying problem? It appears to me that the results of the work point less to the fact that it's a decision between programmatic or neural policy but about the domain knowledge introduced into either system.
> >
> > **3 / 4:** I thank the authors, my concerns regarding these have been addressed. I had misunderstood the setup of the experiments in regards to the motivation of algorithm choice.
> >
> > **5:** I thank the authors for their discussion and agree with their comments. I believe the statement, as written in the paper, could benefit from more context provided in this answer. Furthermore, it would be worthwhile to establish this less as a limitation of neural networks but of a particular modelling choice since parameter sharing techniques, e.g. established in graph neural networks, can directly generalize/ scale to larger input sizes without increasing the number of parameters.
> >
> > ---
> >
> > I will increase my score to weak reject as I consider the work to be borderline in its current state. I am happy to discuss in further discussion regarding my concerns on significance and originality of this work, since these are my key reasons to advocate for rejection at this stage.

---

> > > ### Author Response · Authors · 2026-04-02
> > >
> > > Thank you for taking the time to read our rebuttal and for asking us follow-up questions.
> > >
> > > We will start by explaining where we agree with the reviewer, and then move to the parts that need clarification.
> > >
> > > > Maybe the summary should be that domain knowledge through informed design of programmatic representations or through reward/ feature engineering for neural approaches can boost generalization, independent of whether programmatic or neural policies are being deployed.
> > >
> > > We agree with the above, and this does not conflict with what we stated earlier in our rebuttal, as explained below.
> > >
> > > Let's analyze the paragraph above from bottom to top.
> > >
> > > > independent of whether programmatic or neural policies
> > >
> > > We strongly agree that there shouldn't really be a divide between programmatic and neural. If we leave the engineering cost aside, what matters are the two properties we discussed in our paper: expressivity and discoverability. As long as you can attain both, the representation can be either programmatic or neural.
> > >
> > > > through reward/ feature engineering for neural approaches can boost generalization
> > >
> > > This touches on discoverability. In the experiments we analyzed, both neural and programmatic representations encoded programs as chains of if-then-else statements. This led us to conclude that the neural and programmatic representations used in previous work were all expressive, and that the issue boiled down to whether they were discoverable. Once we controlled for discoverability, both neural and programmatic generalized OOD.
> > >
> > > These tricks wouldn't help with OOD generalization if the representation weren't expressive. We provide an example of this in our paper: in NetHack, the agent needs to solve nested subproblems, which requires a stack. If the representation doesn't support stacks, then there is no reward-shaping trick that will enable OOD generalization. The common approach of "let's collect more data" will never truly work because the representation is not expressive.
> > >
> > > > informed design of programmatic representations [...] can boost generalization
> > >
> > > This touches on expressivity, as at design time we choose the computational constructs a given language will offer, and we agree with it.
> > >
> > > > Maybe the summary should be that domain knowledge [...]
> > >
> > > Domain knowledge is important for addressing the tension between discoverability and expressivity. Once we know which type of computation is required to solve a problem (e.g., stack in the nested subproblem task), we can focus on choosing a representation that satisfies the computational constraints while remaining as discoverable as possible. For example, one could use the models in the Neural Turing Machines line of work. They are always expressive because they are universal. However, in practice, it is difficult to train them, so they will not offer discoverability. To achieve a representation that supports discoverability, we might have to reduce expressivity by leveraging domain knowledge (e.g., in the nested subproblem task, we know we need a stack; this is the kind of domain knowledge we need).
> > >
> > > In general, we argue that future work in representation learning should consider the tension between these two properties: discoverability and expressivity. The dichotomy between neural and programmatic isn't as important as these properties, as our experiments suggest.
> > >
> > > Given our agreement with the reviewer's statements, now let's discuss what needs to be clarified:
> > >
> > > > but one could equally read from the results in Torcs and Karel that neural networks can be beneficial over often more complex programmatic approaches even when specific types of computation are required.
> > >
> > > If Karel and Torcs require types of computation (think of the nested subproblem task that requires a stack) that a neural network cannot deliver, then it is worth the trouble to design a programmatic representation that satisfies the problem's requirements.
> > >
> > > > If one knows which computation is required, through feature engineering or reward shaping based on this domain knowledge, neural policies are shown to perform just as well or better than programmatic policies.
> > >
> > > Tricks such as feature engineering and reward shaping can help with discoverability, but not expressivity. If the neural model cannot perform the computation you need to generalize OOD, no training trick or amount of data will make it generalize OOD.
> > >
> > > We understand the reviewer's concern regarding significance, as it is challenging to evaluate a paper's significance at this stage. From our perspective, we learned a lot from our submission, and it is already shaping our research agenda on representation learning. Instead of creating a divide between neural and programmatic approaches, we are thinking in terms of expressiveness and discoverability, with the grand goal of inventing representations that are universal and yet discoverable. The reviewer is asking great questions, and this is what we were hoping to see with our work.

---

### Official Review · Reviewer_BgU1 · 2026-03-13

**Soundness:** 3
**Presentation:** 3
**Significance:** 2
**Originality:** 3
**Overall Recommendation:** 3
**Confidence:** 3

**Summary:**

This paper revisits a common claim that programmatic policies inherently possess superior out-of-distribution (OOD) generalization compared to neural policies in reinforcement learning. By re-examining three widely used benchmarks (TORCS, Karel, and Parking), the authors show that the previously observed performance gap is primarily due to uncontrolled experimental factors—such as reward function design and observation density—rather than intrinsic representational differences. The study introduces a conceptual framework based on "expressivity" and "discoverability," arguing that when training conditions are adjusted to facilitate the discovery of generalizing solutions (e.g., using cautious rewards or sparse observations), neural networks can match or even exceed the OOD performance of domain-specific programmatic representations.

**Compliance With Llm Reviewing Policy:**

Affirmed.

**Key Questions For Authors:**

1. The paper hinges on the distinction between "Expressivity" and "Discoverability." While the experiments intuitively support this, could the authors provide a more formal or quantitative definition of "Discoverability"? For instance, is there a way to measure the "gradient density" or the volume of the solution space that leads to generalizing behaviors in a neural weight space versus a given DSL?
2. In the TORCS and Parking benchmarks, you introduce "cautious" reward functions to penalize risky behavior. How sensitive is the OOD performance to the specific weighting of these penalties? If the penalty is slightly too low, does the neural network immediately revert to high-reward/low-generalization shortcuts? It would be valuable to see a sensitivity analysis (e.g., a Pareto frontier) between training reward and OOD robustness.

**Limitations:**

Yes

**Strengths And Weaknesses:**

Strengths:
1. The paper identifies a significant "confounding variable" in prior research. It precisely points out that the reported superiority of programmatic policies (e.g., in the TORCS and Karel benchmarks) often stems from unequal training conditions—such as different reward functions or observation spaces—rather than the symbolic nature of the representation itself.
2. The authors do not limit their investigation to a single domain. By replicating and modifying experiments across TORCS (continuous control), Karel (grid-world logic), and Parking (navigation), they provide robust evidence that their findings are generalizable across different types of RL tasks.
3. Instead of just deconstructing prior work, the paper provides concrete solutions. For instance, demonstrating that "Cautious RL" (adjusting reward weights to penalize risk) and "Sparse Observations" (using local sensors instead of global grids) can enable neural networks to achieve OOD performance comparable to programmatic policies.
Weaknesses:
1. A primary motivation for using programmatic policies in the literature is not just OOD generalization, but also their inherent interpretability and the ability to perform formal verification. The paper focuses almost exclusively on performance metrics, potentially downplaying the other structural advantages that programmatic representations offer over "black-box" neural networks.
2. While the concept of "discoverability" is intuitive and well-supported by the experiments, it remains largely qualitative. The paper lacks a more formal or mathematical definition of how the "density" of generalizing solutions in the search space differs between DSLs and neural weight spaces.
3. The benchmarks used (Karel, TORCS) involve relatively simple logic. It remains an open question whether neural networks can still match programmatic policies in tasks requiring extremely deep nested loops or complex recursive logic, where the inductive bias of a DSL might be even more dominant.

---

> ### Author Rebuttal · Authors · 2026-03-26
>
> Thank you for your feedback. We hope to address the points you raised in your review.
>
> **Comment:** Downplaying other structural advantages of programmatic representations.
>
> Our work is about OOD generalization, and that is where we focus our attention in the experiments and discussion. However, we do not downplay other advantages of programmatic representations. In fact, in the last paragraph of Section 6, we recognize them. We agree that there are many other reasons for using programmatic representations in practice. We would be happy to highlight these advantages more prominently in the paper's introduction.
>
> **Question:** The paper hinges on the distinction between "Expressivity" and "Discoverability." While the experiments intuitively support this, could the authors provide a more formal or quantitative definition of "Discoverability"? For instance, is there a way to measure the "gradient density" or the volume of the solution space that leads to generalizing behaviors in a neural weight space versus a given DSL?
>
> This is a wonderful question, and unfortunately, we don't have a good answer to it. We hope our work will raise similar questions that could help advance the field.
>
> **Suggestion:** In the TORCS and Parking benchmarks, you introduce "cautious" reward functions to penalize risky behavior. How sensitive is the OOD performance to the specific weighting of these penalties? If the penalty is slightly too low, does the neural network immediately revert to high-reward/low-generalization shortcuts? It would be valuable to see a sensitivity analysis (e.g., a Pareto frontier) between training reward and OOD robustness.
>
> Yes, there is a threshold from more cautious to less cautious that, once crossed, the model goes from generalizing to not generalizing. This happens due to unsafe decisions, such as speeding on a section of the track with sharp turns. We will include such an analysis in the paper. Thank you for your suggestion.
>
> **Point 6:**
>
> This point raises an important question about discoverability: what can we find with gradient descent versus discrete optimization algorithms used in program synthesis? We believe discoverability is challenging to control for in practice, but this doesn't make it any less important. We hope future work will investigate questions like this one. Thank you for bringing it up.

---

> > ### Author Rebuttal · Reviewer_BgU1 · 2026-04-04
> >
> > Thanks for the rebuttal. I appreciate the clarifications, but my main concerns are not sufficiently resolved, so my score remains unchanged.

---

> > > ### Author Response · Authors · 2026-04-04
> > >
> > > We thank the reviewer for taking the time to read our rebuttal and acknowledge it.
> > >
> > > However, from the reviewer's response, it isn't clear to us what their "main concerns" are that we haven't resolved.
> > >
> > > The initial response contained three weaknesses, which we take as the reviewer's concerns (points 4, 5, and 6 in the review).
> > >
> > > 1. **Point 4** related to something that needed clarification. Specifically, the reviewer thought we were downplaying other strengths of programmatic representations. We then clarified that we don't downplay its other strengths, as we acknowledge all of them in Section 6 of our submission.
> > > 2. **Point 5** makes an interesting suggestion about using solution density as a proxy for discoverability. While this is an interesting idea, it sits outside of the scope of our paper. We are happy to see suggestions like this as they relate to the concepts of our work and show how future papers might build on ours.
> > > 3. In **Point 6**, the reviewer hypothesizes that if the program's structure is more complex than that required in previous work, programmatic representations could outperform neural ones. This hypothesis would have to be tested before we draw conclusions. In our paper, we use the benchmark problems from the papers we reevaluate, but we agree that considering other settings could be interesting for future work.
> > >
> > > In summary, the reviewer raised 3 weaknesses. The first point (point 4) was clarified in our rebuttal. The other two are interesting suggestions for future work, as the questions raised aren't central to our paper's objective, which was to reevaluate existing claims. We see the suggestions in 5 and 6 as ideas that branch out from our work, and we were excited to read about them.
> > >
> > > We kindly ask the reviewer to explain their remaining concerns. Unfortunately, we can't respond to them anymore because this is our last allowed message on this thread, but we would love to know what we are still missing to get our work above the acceptance bar.

---

### Official Review · Reviewer_hH9A · 2026-03-13

**Soundness:** 3
**Presentation:** 3
**Significance:** 2
**Originality:** 2
**Overall Recommendation:** 2
**Confidence:** 3

**Summary:**

The paper revisits the generalization claims of programmatic RL policies. The authors revisit the experiments in the proposed methods, NDPS, LEAPS, PSM, where they had originally shown that programmatic policies are more generalizable than neural policies. The authors argue that as long as the neural policies are expressive enough and the search algorithm can discover the required policy, neural policies are as generalizable as the programmatic policies.

**Compliance With Llm Reviewing Policy:**

Affirmed.

**Key Questions For Authors:**

Please look at the weaknesses.

**Limitations:**

The limitation of the evaluation is mostly the limited set of analysis. Only 3 comparatively older algorithms and simpler benchmarks chosen.

**Strengths And Weaknesses:**

Strengths

(1) Revisiting these claims is useful as misconceptions regarding neural policies is cleared.

(2) The authors discuss when programmatic policies are still useful even though neural policies can have similar OOD generalization. This discussion is useful and helpful for the continued research in programmatic RL.

Weaknesses

(1) I am not very well versed in programmatic RL literature but the papers and algorithms the authors discuss are from 2018, 2019, 2020 and 2021. And revisit the experiments in these methods. Are there no recent work in programmatic RL (in more complex domains) that talk about the generalization capacities. There are some methods vaguely mentioned towards the end but then are all the papers claiming better generalization for programmatic policies incorrect?

(2) There are three representative algorithms each on a different benchmark. While the language of the program are different, can they not run on the other benchmarks with the corresponding language. For instance, using the CFG of KAREL, can NDPS be trained on KAREL? Because they have not been done so in the paper.

While the paper produces an interesting insight, the results are not unsurprising. Programmatic policies have a very small search space with lot of inductive biases (due to the program) so they are less sensitive to spurious correlations. Neural policies have massive parameter space so the search is difficult. A careful construction of the neural policies with carefully constructed reward functions and architectures can mitigate some of the spurious correlations. Moreover, the domains and experiments chosen by the authors are very simple. They could have chosen some robotics domains where programmatic policies has been applied [1, 2, 3].

[1]: Holtz, et. al., Robot Action Selection Learning via Layered Dimension Informed Program Synthesis, CoRL 2020

[2]: Holtz, et. al., Iterative Program Synthesis for Adaptable Social Navigation, IROS 2021

[3]: Xin, Jimmy, et al. "Programmatic imitation learning from unlabeled and noisy demonstrations." IEEE Robotics and Automation Letters

---

> ### Author Rebuttal · Authors · 2026-03-26
>
> Thank you for your feedback.
>
> **1. Are all the papers claiming better generalization for programmatic policies incorrect?**
>
> What we refer to in Section 6 (Relation to Other Works) is that generalization claims could be incorrect if the neural network represents the same space of programs given by the language used in synthesis. We carefully analyzed the claims of three papers from the literature. The most recent work with similar claims, mentioned in Section 6, is by Qiu \& Zhu (2022), which also used RL benchmarks that can be trained with models similar to those used in our study.
>
> While it is difficult to say anything about *all papers*, in general, the claims are likely correct if the underlying problem requires computational resources that a neural network cannot provide (e.g., growing memory requirements) but that a programmatic representation can.
>
> **2. Using the CFG of KAREL, can NDPS be trained on KAREL? Because they have not been done so in the paper.**
>
> Yes, with some adaptations, the synthesis process used in one domain may be applicable to other domains. However, this would defeat the purpose of our experiments, as we want to evaluate the approaches as presented in their original papers. Otherwise, our experiments could be flagged as wrong for not using the implementations presented in their original papers.
>
> **3. Results are unsurprising**
>
> While it is difficult to argue whether a result is surprising, we show that papers from different groups didn't control for biases in neural representations in their experiments. In that sense, if the result appears unsurprising now, we believe this conclusion emerges only in hindsight.
>
> **4. Why not robotics papers?**
>
> None of the robotics papers mentioned [1, 2, 3] make OOD generalization claims similar to those we investigate. In fact, we are not aware of any robotics papers that make such claims and thus meet our study's inclusion criteria. All three papers from Joydeep Biswas' group that the reviewer mentions evaluate forms of generalization (policy repair in [1, 2] and robustness to noise in [3]). They don't directly compare fixed neural and programmatic solutions under distribution shift, as the papers we evaluated do. The three papers do point to other dimensions in which programmatic and neural policies could be compared, though. Although they differ from the OOD generalization we consider, they are also important and will be included in our Section 6, where we discuss other advantages that programmatic representations have over neural ones.

---

> > ### Author Rebuttal · Reviewer_hH9A · 2026-04-04
> >
> > Thank you for addressing my concerns. Some of my concerns still remain though. If the authors mention that "The most recent work with similar claims, mentioned in Section 6, is by Qiu & Zhu (2022)", and the algorithms discussed in the paper are from 2018 to 2021, does this mean that the community implicitly agreed that these claims are not entirely correct? This ties to my concern about the results not being surprising.
> >
> > The paper questions the claims made by a few programmatic RL papers some years ago but if recent papers no longer address these claims (as said by the authors, the most recent paper being one from 2022 and the recent robotics papers on programmatic RL not making these claims) implies that the community already agrees that these claims are not entire correct which reduces the impact of this paper.

---

> > > ### Author Response · Authors · 2026-04-04
> > >
> > > Thank you for reading our response and taking the time to explain the remaining concern you have regarding our work.
> > >
> > > >The paper questions the claims made by a few programmatic RL papers some years ago but if recent papers no longer address these claims (as said by the authors, the most recent paper being one from 2022 and the recent robotics papers on programmatic RL not making these claims) implies that the community already agrees that these claims are not entire correct which reduces the impact of this paper.
> > >
> > > No, the community has not implicitly agreed that the claims we revisit are incorrect, and we can prove this (see the next paragraph). We indeed observe that no more papers have been published in the past few years making similar claims. This is likely due to how the ML community reviews papers: we don't normally accept experiments that repeat what has already been published; reviewers would push back, saying that these generalization claims have already been made by the very papers we study.
> > >
> > > Instead of looking for papers that make similar claims, the right way to see how the community understands this topic is to examine how the papers we studied are cited. These three papers remain widely cited for the exact claim we study in our work. Please see the following excerpts of recent papers:
> > >
> > > 1. "Aiming to produce reinforcement learning (RL) policies that are human-interpretable and can generalize better to novel scenarios, Trivedi et al. (2021) present a method (LEAPS)" [Abstract of Liu et al. Hierarchical Programmatic Reinforcement Learning via Learning to Compose Programs **(ICML 2023)**]
> > > 2. "Previous work showed that due to the inductive bias of the language in which such policies are written, they tend to generalize better to unseen scenarios (Inala et al., 2020; Trivedi et al., 2021)" [Introduction of Carvalho et al. Reclaiming the Source of Programmatic Policies: Programmatic versus Latent Spaces **(ICLR 2024)**]
> > > 3. "Programmatic representations of policies for solving Markov Decision-Processes (MDPs) offer advantages over neural representations, such as the ability to better generalize to similar but different scenarios than those used in training (Inala et al., 2020)" [Introduction of Moraes and Lelis. Searching for Programmatic Policies in Semantic Spaces **(IJCAI 2024)**]
> > > 4. "[...] program policies are shown to be able to capture high-level task-solving ideas, allowing for generalizing to a wide
> > > range of task variants (Trivedi et al., 2021)" [Introduction of Liu et al. Synthesizing Programmatic Reinforcement Learning Policies with Large Language Model Guided Search **(ICLR 2025)**]
> > > 5. "Another desirable property is generalizability: programmatic policies are expected to generalize better, as it was argued in the original papers (Bastani, Pu, and Solar-Lezama 2018; Inala et al. 2020; Trivedi et al. 2021). [Section 6 of Shabadi, Fijalkow, and Matricon. Programmatic Reinforcement Learning: Navigating Gridworlds **(arXiv 2024)**]"
> > >
> > > These recent citations clearly show that the community has not implicitly agreed that the claims we challenge in our work are incorrect. Our paper invites all these researchers to reconsider the assumption that programmatic representations are inherently superior to neural representations for OOD generalization.

---

### Decision · Program_Chairs · 2026-04-30

**Decision:**

Accept (regular)

**Comment:**

This paper re-evaluates the prevailing claim that programmatic policies inherently possess superior out-of-distribution generalization compared to neural policies in reinforcement learning. By re-examining the TORCS, Karel, and Parking benchmarks, the authors demonstrate that neural networks can match or exceed programmatic performance when experimental factors like reward functions and observation density are controlled. The primary strengths include identifying significant confounding variables in prior research and introducing a conceptual framework based on expressivity and discoverability. However, reviewers noted weaknesses regarding the limited novelty of the technical modifications, the relatively simple benchmarks used, and a lack of formal mathematical definitions for key qualitative concepts.

The authors' rebuttal addressed concerns about the significance of their work by providing evidence that the community continues to widely cite the challenged papers for their generalization claims. They also clarified that their focus was strictly on out-of-distribution generalization rather than other structural benefits of programs like interpretability. While the authors successfully convinced one reviewer to raise their score by clarifying how their findings advance the field's perspective, some reviewers remained unsatisfied. Remaining concerns include the lack of a quantitative metric for discoverability and the possibility that the results may seem unsurprising in hindsight.

I am slightly leaning toward accepting this paper because it successfully challenges a widely held assumption in the reinforcement learning community by disentangling representational benefits from experimental confounds. This is a weak accept, as the paper's contribution is primarily empirical and critical rather than methodological, but it provides a necessary course correction for future research.